# Investigation of In Vitro Susceptibility and Resistance Mechanisms in Skin Pathogens: Perspectives for Fluoroquinolone Therapy in Canine Pyoderma

**DOI:** 10.3390/antibiotics11091204

**Published:** 2022-09-06

**Authors:** Stefano Azzariti, Ross Bond, Anette Loeffler, Flavia Zendri, Dorina Timofte, Yu-Mei Chang, Ludovic Pelligand

**Affiliations:** 1Department of Comparative Biomedical Sciences, Royal Veterinary College, Hawkshead Lane, North Mymms, Hatfield AL9 7TA, UK; 2Department of Clinical Sciences and Services, Royal Veterinary College, Hawkshead Lane, North Mymms, Hatfield AL9 7TA, UK; 3Institute of Infection, Veterinary and Ecological Sciences, Department of Veterinary Anatomy, Physiology and Pathology, Veterinary Microbiology Diagnostic, University of Liverpool Leahurst Campus, Neston CH64 7TE, UK; 4Research Support Office, Royal Veterinary College, University of London, London NW1 0TU, UK

**Keywords:** canine pyoderma, staphylococci, *S. pseudintermedius*, *S. aureus*, *E. coli*, antimicrobial resistance, methicillin-resistance

## Abstract

Fluoroquinolones (FQ) are commonly used in dogs with bacterial skin infections. Their use as first choice, along with the increased incidence of FQ-resistance, represents a risk to animal and public health. Our study determined minimum inhibitory (MIC) and bactericidal (MBC) concentrations of five FQs in *Staphylococcus aureus*, *Staphylococcus pseudintermedius,* and *Escherichia coli,* together with FQ-resistance mechanisms. MICs, efflux pump (EP) overexpression and MBCs were measured in 249 skin infection isolates following CLSI guidelines (CLSI VET01-A4, CLSI M26-A). Chromosomal and plasmid-mediated resistance genes were investigated after DNA extraction and sequencing. FQ-resistance was detected in 10% of methicillin-susceptible (MS), 90% of methicillin-resistant (MR) staphylococci and in 36% of *E. coli*. Bactericidal effect was observed except in 50% of MRSA/P for ciprofloxacin and in 20% of MRSPs for enrofloxacin. Highest MICs were associated with double mutation in *gyrA* (Ser83Leu + Asp87Asn), efflux pumps and three PMQR genes in *E. coli*, and *grlA* (Ser80Phe + Glu84Lys) in *S. aureus*. EP overexpression was high among *E. coli* (96%), low in *S. aureus* (1%) and absent in *S. pseudintermedius*. Pradofloxacin and moxifloxacin showed low MICs with bactericidal effect. Since in vitro FQ resistance was associated with MR, FQ use should be prudently guided by susceptibility testing.

## 1. Introduction

Canine pyoderma, a cutaneous pyogenic bacterial infection [1], represents one of the most common skin diseases in dogs [2], and is caused, in almost 90% of cases, by *Staphylococcus pseudintermedius* [3]. Other less-frequently isolated pathogens include *Escherichia coli*, *Pseudomonas aeruginosa*, and *Streptococcus* spp. [1,4]. *S. aureus* is rarely found on canine skin [5], and the majority of isolates are human-related [6]. However, the increased incidence of infections caused by multidrug-resistant (MDR) and methicillin-resistant (MR) staphylococci (MRSP and MRSA) in dogs is a public health concern [7,8], along with the increased frequency of infections caused by *E. coli* [9,10,11].

Fluoroquinolones (FQ) are efficacious and licensed for the systemic treatment of bacterial skin infections in dogs [12,13,14,15,16], and their use is advocated in deep pyoderma [17], or widespread and severe superficial pyoderma when antimicrobial susceptibility testing (AST) results are compatible.

FQs have a broad spectrum of activity and are listed as second choice antimicrobials [18], or also classified as category B “prudent” by the European Medicine Agency (EMA), following WHO guidelines on critically important antimicrobials (CIA) in human medicine [19,20]. Their use should be prudently limited to clinical cases where first-line antibiotics have been ineffective [13,21,22], to reduce the risk of antimicrobial resistance both in commensal and pathogenic bacteria [23]. FQ-resistance usually arises in a stepwise manner, when bacteria are sequentially exposed to the drug, and is mediated by mutations of genes encoding for DNA gyrase (*gyrA* and *gyrB*), and topoisomerase IV (*parC* and *parE in* Gram-negative, or *grlA* and *grlB* in Gram-positive). As a result, FQ binding affinity to the topoisomerases is reduced due to alteration of the target protein structure secondary to amino acid substitution [24,25]. Other resistance mechanisms described include: (1) decreased permeability of bacterial cell wall due to down-regulation and under-expression of outer membrane porins, (2) overexpression of efflux pumps [26,27,28,29,30,31,32], (3) resistance conferred by plasmid-mediated quinolone resistance (PMQR) genes, which includes FQ degradation (*aac(6′)-Ib-cr*) [33], efflux pumps (*qepA*) [28,29] or disruption of the interaction with FQs by binding to topoisomerases (Qnr family) [34]. Efflux pump upregulation can cause a 4–8-fold increase in MIC. However, major contributions in decreased susceptibility are caused by multiple mutations (up to 128-fold) and, in Gram-negatives by PMQR genes [35].

Previous studies have investigated the susceptibility and prevalence of resistance mechanisms among staphylococci and *E. coli* of canine origin [6,31,36,37,38,39,40,41,42,43,44,45,46], but limited information exists on the bactericidal activity of isolates collected from canine pyoderma or wound infection cases.

Early veterinary FQs can bind both topoisomerases, conferring bactericidal activity. However, their primary targets are specific for bacterial species: DNA gyrase for Gram-negative and Topoisomerase IV for Gram-positive bacteria, respectively [47,48,49]. In comparison, newer generation pradofloxacin and moxifloxacin may represent an advantage in reducing the likelihood of resistance development as the drugs target both bacterial topoisomerases with increased affinity, conferring low minimum inhibitory concentrations (MIC) and mutant prevention concentrations (MPC) [50,51].

Pradofloxacin, licensed for dogs in Europe in oral formulations [52], has a very similar molecular structure to moxifloxacin, a drug licensed for human use [53]. Silley et al. [54], showed that pradofloxacin exhibits bactericidal activity with regards to minimum bactericidal concentration (MBC), the minimum concentration that kills 99.9% of bacteria. The MBC values were within two MIC doubling dilutions against 90% of selected isolates from unspecified animal species and body sites, but to date no veterinary studies have compared MBC of pradofloxacin with other FQs in isolates specifically from canine skin.

Here we compared MICs and MBCs distributions of selected veterinary and human fluoroquinolones in *S. pseudintermedius*, *S. aureus,* and *E. coli* isolates from canine pyoderma or skin wound infections, with a particular focus on the correlation between methicillin and FQ resistance. It was hypothesized that MICs and MBCs distributions of clinical isolates differed between FQs, and MR in staphylococci was associated with increased MICs and MBCs that predicted FQ-resistance. We also evaluated chromosomal mutations, efflux pump overexpression and the presence of PMQR genes amongst FQ-resistant skin pathogens, with the objective to correlate the resistance mechanisms with susceptibility. We hypothesized that the presence of multiple chromosomal mutations, efflux pumps, and PMQR genes were associated with increased MICs.

## 2. Results

MICs, MBCs values, MBC/MIC ratios and their respective ranges are shown in Table 1. MICs graphical distributions are shown in Figure 1. Chromosomal mutations, PMQR genes, and efflux pump overexpression detection are represented in Table 2. Appendix A comprise QC growth ranges (Appendix A), clinical breakpoints (Appendix A), graphical MBCs distributions (Appendix A) and statistical analyses comparing MICs (Appendix A) and MBCs (Appendix A).

### 2.1. Minimum Inhibitory Concentration

MICs distributions were bimodal for the three species (Figure 1). In staphylococci, MS and MR resistant isolates showed different modes within each distribution.

*S. aureus*: 31 out of 34 MSSA (91.2%) and 2 out of 45 MRSA (4.4%) were FQ-susceptible. Among the resistant isolates, two MSSA and two MRSA were resistant to all FQs except for pradofloxacin (MIC 1 µg/mL). Pradofloxacin and moxifloxacin had lower MICs (*p* < 0.0001) both for MRSA and MSSA (Appendix A), compared with enrofloxacin, marbofloxacin and ciprofloxacin. No statistical difference (*p* > 0.05) was observed between marbofloxacin and ciprofloxacin (*p* = 0.87) and between pradofloxacin and moxifloxacin (*p* = 0.34). MICs for MRSA were higher than for MSSA (*p* < 0.0001) for all FQs tested.

*S. pseudintermedius*: 50 out of 53 MSSP (94.3%) and 3 out of 52 MRSP (5.8%) were FQ- susceptible. Among the resistant isolates, one MSSP was resistant to all FQs except for pradofloxacin and moxifloxacin (MIC 0.5 µg/mL with both drugs) and two MRSP were susceptible to pradofloxacin (1 µg/mL).

MICs were lower for each FQ tested in MSSP compared with MRSP (*p* < 0.0001) and lower MICs were observed between both pradofloxacin, moxifloxacin (*p* < 0.0001) and the other FQs tested.

*E. coli*: 41 out of 65 isolates (63%) were FQ-susceptible except for moxifloxacin (36 out of 65, 55%). Highly significant differences in MICs and MBCs distribution between fluoroquinolones (*p* < 0.0001) were observed, and pradofloxacin showed the lowest MICs and MBC (*p* < 0.0001) when compared with the other FQs tested.

### 2.2. Minimum Bactericidal Concentration

MBCs distributions were bimodal for the three species (Appendix A). In staphylococci, MS and MR isolates showed different modes within each distribution.

*S. aureus*: bactericidal effect was observed in all the isolates with an MBC within four two-fold dilutions of the MIC value, except for enrofloxacin (1 out of 34 MSSA, 2.9% and 1 out of 45 MRSA, 2.2%), ciprofloxacin (22 out of 45 MRSA, 48.8%) and moxifloxacin (1 out of 45 MRSA, 2.2%). Pradofloxacin and moxifloxacin had lower MBC values (Appendix A), both for MSSA and MRSA, in comparison with the other FQs (*p* < 0.0001) and higher values were associated with methicillin resistant isolates (*p* < 0.0001). MBCs for MRSA were higher than MSSA (*p* < 0.0001) for all FQs tested.

*S. pseudintermedius*: pradofloxacin and moxifloxacin had lower MBCs (*p* < 0.0001) compared to the other FQs tested. All the isolates showed MBCs within four doubling dilutions except for enrofloxacin (10 out of 52 MRSP, 19.2%), ciprofloxacin (29 out of 52 MRSP, 55.7%).

*E. coli*: pradofloxacin showed the lowest MBCs (*p* < 0.0001) when compared with the other FQs tested. MBC/MIC ratios were within four doubling dilutions only with pradofloxacin and moxifloxacin.

### 2.3. Resistance Mechanism Detection

*S. aureus:* in five selected resistant isolates (two MSSA and three MRSA), *gyrA* showed either single (Ser84Leu, three out of six isolates) or double (Ser84Leu + Gly90Cys, two out of six isolates) chromosomal mutations. A single mutation (ser80Phe) was observed on *grlA* (Table 2) in four out of six isolates with pradofloxacin MIC ≤ 4 µg/mL whereas two mutations were observed in one out of six isolates with high pradofloxacin MIC (32 µg/mL). No mutations were detected in one isolate showing efflux pump overexpression, which represented 1% (1 out of 79) of the resistant isolates.

*S. pseudintermedius*: in six selected resistant isolates (3 MSSA and 3 MRSA), single mutations were found on *gyrA* (Ser84Leu, five out of six, and Glu88Gly, 1 out of six) and on *grlA* (Ser80Ile, Ser80Arg, Asp84Asn; four, one and one out of six, respectively) (Table 2) both in isolates with intermediate susceptibility (MIC < 2 µg/mL) or low resistance (MIC ≥ 2 µg/mL) to pradofloxacin. None of the isolates showed efflux pump overexpression.

*E. coli*: all FQ-resistant selected isolates showed double mutations on *gyrA* (Ser83Leu + Asp87Asn) and single (Ser80Ile) mutation on *parC.* Double mutations (Ser80Ile + Ala108Val) on *parC* were observed in three out of six isolates (Table 2).

MICs in the presence of an efflux pump inhibitor (EPI) Phenylalanine-arginine beta-naphthylamide (PAβN) were 4- to 32-fold (median 8-fold) lower (*p* < 0.0001) than pradofloxacin MICs and 23 out of 24 (96%) of the resistant isolates showed efflux pump overexpression.

Eleven out of twenty-four (46%) resistant *E. coli* carried PMQR genes: 8 out of 24 (33%) were *aac-(6′)-lb-cr,* 3 out of 24 (12.5%) *qnrS* and 2 out of 24 (8.3%) *qnrB*, whereas the *qnrA* gene was not detected. Two isolates with high MICs were associated double PMQR genes, the first one with *qnrS* and *aac-(6′)-lb-cr,* and the second one with *qnrS* and *qnrB*. *qepA* and *oqxAB* genes were not detected.

## 3. Discussion

Our data indicate that pradofloxacin and moxifloxacin overall show low MICs with bactericidal effect both in staphylococci and *E. coli*, compared to earlier generation veterinary and human FQs. The results are compatible with and expand those previously published on MICs [31,38,40,42,51,55,56,57], in isolates collected from canine skin infections. However, this is the first veterinary study that examined a substantial number of MS and MR staphylococci in canine pyoderma. The percentages of FQ-resistances were high among MR staphylococci (90%) and *E. coli* (36–40%), in line with previous studies, where resistance was also detected among four non-β-lactam classes [40,58,59,60]. Since the MIC data are obtained from different countries and periods of time, future analysis is needed to investigate the differences in terms of antibiotic pressure and the related resistance development between old and new generation FQs.

This is the first report on MBC of veterinary and human FQs in canine skin isolates. Previous studies showed that the MBC of ciprofloxacin, moxifloxacin, and pradofloxacin in MS and MR *S. aureus* [61], and pradofloxacin in *S. pseudintermedius* [54], was within four doubling dilutions of the MIC in all isolates. In *E. coli*, ciprofloxacin [62], and pradofloxacin [54], showed bactericidal effect in almost all isolates. Our study confirmed the high bactericidal effect in *E. coli* collected from canine pyoderma and extended to other veterinary and human FQs. However, in staphylococci no bactericidal effect (MBC/MIC ratio > four) was observed for ciprofloxacin in 50% of MRSA and MRSP and in 20% of enrofloxacin in MRSP. A comparison of the MBC_90_ of ciprofloxacin in MRSA in our study with results from a similar study by Smith and Eng [63], reflects an alarming increase of more than 512-fold over a 30-year period. 

Mutations on *S. pseudintermedius* isolates were found in codons *gyrA*84 and *grlA*80 in all selected FQ-resistant isolates as previously reported [45,64], except in one isolate with intermediate/increased susceptibility to pradofloxacin (0.5 µg/mL) with a single mutation on codon 88 (Glu88GLy). The same mutation was previously identified by Descloux et al. [45], in one FQ-resistant isolate with an MIC of enrofloxacin at the breakpoint (4 µg/mL) and one isolate from Japan [65], with intermediate susceptibility to ofloxacin and resistance to enrofloxacin and levofloxacin. In *S. aureus*, mutations on *gyrA* (Ser84Leu) and *grlA* (Ser80Phe) were identified both in isolates with intermediate susceptibility and resistance to pradofloxacin. However, the presence of an additional mutation on *gyrA* (Gly90Cys), together with *grlA* (Glu84Lys) as reported by Hiasa et al. [66], conferred high resistance to pradofloxacin (32 µg/mL) and moxifloxacin (16 µg/mL). Further screening is therefore necessary to understand the molecular mechanisms that confer low resistance profiles in pradofloxacin and moxifloxacin when compared to higher resistant MICs in early generation FQs.

All *E. coli* had two mutations on *gyrA* (Ser83Leu, Asp87Asn) and one mutation on *parC* (Ser80Ile). However, additional mutations on *parC* (Ala108Val, Glu84Val) were screened in isolates with both moderate and, if associated with plasmid genes, high resistance profiles.

Mutations were found to be within the quinolone resistance determining region (QRDR), which are in the proximity of the FQ-binding sites of the primary targets (Tyr122 in *E. coli gyrA* and Tyr119 in staphylococcal *grlA*). Mutations are known to reduce the hydrogen bond with FQs [67], with the addition of loss in negative charge with mutations in *parC* [68]. However, in silico tools such as molecular docking of FQs into the native and mutated protein are necessary to elucidate the primacy of enzyme targeting in veterinary FQs and their associated reduced resistance emergence.

Resistance conferred by efflux pumps was observed to be highly prevalent in *E. coli* (96%), compared to in *S. aureus* (1%) or absent in *S. pseudintermedius*. Pradofloxacin was the only FQ tested against all three bacterial species to detect efflux pump overexpression. Further research is therefore needed to investigate the susceptibility of other fluoroquinolones. It has been suggested that more lipophilic FQs (such as enrofloxacin, and moxifloxacin) are more readily excreted by efflux pumps in comparison to hydrophilic FQs (such as ciprofloxacin, pradofloxacin, and marbofloxacin) [38,69,70,71,72,73]. In *S. pseudintermedius*, the mechanism was not detected in our study, which is in accordance with previous studies [31,32]. Although efflux pump overexpression was observed in only 1 out of 79 of *S. aureus* isolates, to our knowledge, this is the first report of FQ resistance due to an efflux pump mechanism in *S. aureus* from canine pyoderma. Similar results were obtained by Schmitz et al. [74], with moxifloxacin, where the change in MIC after exposure to reserpine were negligible (1–2 dilutions), compared to ciprofloxacin (1–4 dilutions). In contrast to staphylococci, almost all the *E. coli* isolates showed efflux pump overexpression mechanism, and we identified a higher percentage compared to previous studies on *E. coli* collected from canine otitis or different body sites [37,38].

With regards to PMQR, nearly 50% of FQ-resistant isolates were associated with at least one PMQR gene. Among the genes screened for, *aac-(6′)-lb-cr* (33%) and *QnrS* (12.5%) were the most prevalent, followed by *QnrB* (8.3%) as also demonstrated in other studies [37,38,75], whereas no *QnrA* was found. The presence of *QnrB* together with *QnrS* was associated with high MICs.

Limitations of the study are represented by the lack of veterinary CBPs for moxifloxacin and ciprofloxacin, that were extrapolated from human guidelines. The higher percentage of resistance of moxifloxacin in *E. coli* may be associated with discrepancies between EUCAST and CLSI breakpoints guidelines. As other authors have highlighted, this addresses the need for harmonized guidelines across countries [76]. Moreover, CBPs were not available for *S. pseudintermedius* and were extrapolated from *S. aureus* as adopted by previous investigators [51].

## 4. Materials and Methods

### 4.1. Bacterial Pathogens

A total of 249 isolates from canine skin or wound infections were included in this study. *S. aureus* (*n* = 79, 34 methicillin susceptible [MSSA] and 45 methicillin resistant [MRSA]) and *S. pseudintermedius* (*n* = 105, 53 MSSP and 52 MRSP), were collected from Germany between 2005–2006 and 2010–2011, respectively. *E. coli* (*n* = 65) were collected from the UK (Royal Veterinary College and University of Liverpool) from 2010 and 2017. Species identification was confirmed by the presence of the genes *nucA* [77,78], in staphylococci and *uidA* [79], and *uspA* [80], in *E. coli*. Methicillin resistance in staphylococci was investigated by the presence of the *mecA* gene [81].

### 4.2. Antimicrobial Agents Tested

Enrofloxacin, marbofloxacin, ciprofloxacin, and moxifloxacin were purchased from Merck (Steinheim, Germany). Pradofloxacin was provided by Bayer Animal Health (Monheim, Germany). Stock solutions (1 mg/mL) were dissolved in deionized water, filter sterilized, adjusted for potency according to Clinical and Laboratory Standard Institute guidelines (CLSI VET01-A4, 2013) [82], and stored in darkness at −80 °C for up to one month.

### 4.3. Minimum Inhibitory and Bactericidal Concentrations

MICs were determined by broth microdilution (CLSI VET01-A4) [82]: isolates were recovered from Brain Heart Infusion (BHI, ThermoFisher, Basingstoke, UK) with 25% glycerol at −80 °C, subcultured onto 5% sheep blood (TCS Bioscience, Botolph Claydon, UK) agar plates (BA, Merck, Steinheim, Germany) and incubated at 37 °C for 18–24 h. After incubation, 3–5 colonies were suspended into glass tubes containing phosphate buffer saline (PBS ThermoFisher, Basingstoke, UK). A spectrophotometer (Densichek^®^, Biomérieux, Marcy L’Étoile, France) was used to standardize optical density to 0.5 McFarland, equal to 1–2 × 10^8^ colony forming unit/mL (CFU/mL) and the bacterial suspension was diluted to achieve a final concentration of 5 × 10^5^ CFU/mL in a 96-well MIC plate (Sarstedt^®^, Nümbrecht, Germany) containing two-fold dilution series of the antimicrobial agents. Plates were incubated at 37 °C for 18–24 h and after incubation were manually read against a black paper background; MIC was considered as the lowest concentration where no visible growth was observed for each isolate. MIC_50_ and MIC_90_ (50th and 90th percentiles of the distribution) were calculated as the lowest concentrations that inhibited the growth of 50% and 90% of the isolates, respectively. *S. aureus* ATCC 29213 and *E. coli* ATCC 25922 were included for quality control purposes. Their respective ranges of MICs are shown in Appendix A. Inoculum density and culture purity were confirmed according to EUCAST guidelines (2022) [83].

MICs data were compared with clinical breakpoints published by CLSI guidelines VET01S (2020) [84], for veterinary fluoroquinolones, and CLSI M100 (2022) [85], for human fluoroquinolones. Isolates with “intermediate” susceptibility were considered susceptible with increased exposure as reported by EUCAST. Epidemiological cut-offs (ECOFFs) were also included, if available from datasets, to distinguish between wild- and non-wild-type isolates. Moxifloxacin clinical breakpoints for *E. coli* were obtained from EUCAST guidelines (2022) [86]. For *S. pseudintermedius*, moxifloxacin and ciprofloxacin CBPs were adopted from CLSI for *S. aureus* as previously described in a veterinary study [51]. Clinical breakpoints are listed in Appendix A.

After MIC determination, MBC was measured according to CLSI M26-A “Methods for Determining Bactericidal Activity of Antimicrobial Agents; Approved Guidelines” [87].

MBC_50_ and MBC_90_ (50th and 90th percentiles of the distribution) were calculated as the lowest concentrations that produced at least a 3log_10_ reduction (99.9% bacterial killing) of viable bacterial populations (i.e., <5 × 10^2^ CFU/mL) in antibiotic-treated wells where no visible growth was observed in 50% and 90% of the isolates, respectively. Moreover, drugs were considered bactericidal if the MBC/MIC ratio was ≤ 4.

### 4.4. Efflux Pump Overexpression

Resistant isolates were tested, according to the same MIC method previously described (CLSI VET04), with the addition of EPIs: 20 µg/mL of reserpine and 80 µg/mL of PaβN for *E. coli* were added to MIC plates containing 2-fold pradofloxacin (used as standard FQ) dilutions. Overexpression of efflux pump was detected if the ratio between MIC in the absence and MIC in the presence of EPI (MIC/MIC_EPI_) was ≥4.

### 4.5. Chromosomal Mutations and PMQR Genes

Eighteen FQ-resistant isolates (6 *S. aureus*, 6 *S. pseudintermedius,* and 6 *E. coli*, Table 2) were chosen to represent different resistance profiles based on pradofloxacin MICs (low/susceptible with increased exposure, medium, and high resistance).

DNA was sequenced for the presence of chromosomal mutations on the first subunit of the two topoisomerases (DNA gyrase and topoisomerase IV) targeted by fluoroquinolones, namely *gyrA* and *grlA* (*parC* for *E. coli*).

DNA was extracted from bacterial cells: 5–6 colonies of an overnight culture on sheep blood agar were suspended into Eppendorf tubes containing 1 mL of PBS and centrifuged at 5000 g for 10 min to pellet cells. The supernatant was removed, cell pellet was resuspended in 100 µL of Tris EDTA (TE) buffer and heated at 100 °C for 10 min. For staphylococci, the heated suspension was incubated on ice for 1 min to degrade the cell wall. The final step involved centrifugation at 5000 g for 30 s to pellet cell debris and collect supernatant into sterile Eppendorf tubes.

*GyrA* and *grlA* (*parC*) genes were amplified with polymerase chain reaction (PCR) using previously published protocols [64,88,89]. PCR products were run with electrophoresis on 2% agarose gel together with a positive and a negative control. Results were read with a transilluminator, and results saved on electronic files.

DNA concentration was measured with a fluorometer (Qubit, Invitrogen) and diluted to a standard concentration of 10 ng/mL. Each primer (both forward and reverse) was separately diluted to a standard concentration of 1.3 nmol/µL. Samples were analyzed by Source bioscience (Cambridge) and genetic sequences were aligned and compared with QC isolates with Bioedit 7.2 version.

Moreover, all FQ-resistant *E. coli* were screened for carriage and prevalence of plasmid-mediated quinolone resistance (PMQR) genes: *QnrA, QnrB, QnrS* [90], *aac(6')-lb-cr*, *qepA,* and *oqxAB* [91], genes were amplified by PCR according to previously published methods.

### 4.6. Statistical Analysis

MICs and MBCs data were log_2_ transformed before statistical analysis and a Shapiro–Wilk test was used to assess normality of the distributions. MS and MR staphylococci were considered as two different bacterial types within the same species.

MICs and MBCs distributions were compared between 5 fluoroquinolones within bacterial type (MS, MR staphylococci and *E. coli*) using Anderson Darling and Friedman’s tests. Dunn’s post hoc analysis was carried out for pairwise comparisons.

Wilcoxon signed-rank tests were used to compare the median MICs distributions of pradofloxacin in presence and absence of EPI. Analyses were performed using GraphPad Prism version 9.0 statistical software package (San Diego, CA, USA) with *p* < 0.05 for significance.

## 5. Conclusions

Lowest MICs and MBCs were measured with pradofloxacin and moxifloxacin. The presence of methicillin resistance can predict FQ-resistance in more than 90% of MR staphylococci and in 36% of *E. coli*. Bactericidal effect may not be achieved in MR staphylococci with enrofloxacin and ciprofloxacin.

Since FQ-resistance is multifactorial, further molecular screening of resistance mechanisms and its correlation with antimicrobial susceptibility are necessary.

These findings address the need for prudent use of FQs that should be preventively guided by antimicrobial susceptibility testing prior to systemic treatment.

## Figures and Tables

**Figure 1 antibiotics-11-01204-f001:**
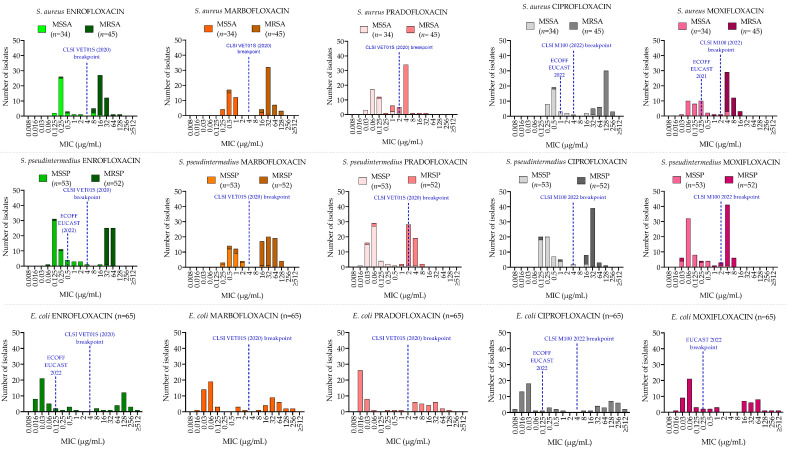
MIC distributions in *S. aureus* (first row), *S. pseudintermedius* (second row) and *E. coli* (third row) of five fluoroquinolones. Blue-dotted lines indicate the clinical breakpoint, the highest MIC value considered susceptible, and the epidemiological cut-off (ECOFF), the MIC value that separates wild-type from non-wild-type bacteria.

**Table 1 antibiotics-11-01204-t001:** MIC, MBC values and MBC/MIC ratios of five fluoroquinolones among a total of 249 isolates of three bacterial species (*S. aureus*, *S. pseudintermedius*, *E. coli*) and their subtypes (MS and MR staphylococci), isolated from canine pyoderma and/or wound infection cases.

Bacterial Type	Fluoroquinolone	MIC (μg/mL)			MBC (μg/mL)			MBC/MIC Ratio *		
		MIC50	MIC90	Range	MBC50	MBC90	Range	MBC50/MIC50	MBC90/MIC90	Range
MSSA (*n* = 34)	ENR	0.25	2	0.125–16	0.25	2	0.125–16	1	2	1–8
	MAR	0.5	1	0.25–32	1	2	0.25–32	1	2	1–4
	PRA	0.06	0.125	0.03–2	0.125	0.25	0.03–4	1	2	1–2
	CIP	0.5	2	0.25–32	0.5	8	0.25–64	1	2	1–4
	MOX	0.125	0.5	0.03–4	0.25	0.5	0.06–4	1	2	1–4
MRSA (*n* = 45)	ENR	16	32	0.25–128	32	64	0.5– ≥512	2	4	1–8
	MAR	32	64	0.5–128	64	64	0.5–128	1	2	1–2
	PRA	4	4	0.125–32	4	8	0.25–32	1	2	1–4
	CIP	128	128	0.5–256	256	≥512	0.5 – ≥512	4	16	1–32
	MOX	4	8	0.125–16	8	16	0.125–32	1	2	1–8
MSSP (*n* = 53)	ENR	0.125	2	0.125–32	0.25	2	0.125–64	1	2	1–4
	MAR	0.25	2	0.25–32	0.5	2	0.25–64	1	2	1–4
	PRA	0.06	0.25	0.016–4	0.06	0.25	0.03–4	1	2	1–8
	CIP	0.25	1	0.125–16	0.25	2	0.125– ≥512	1	2	1–64
	MOX	0.06	0.5	0.03–4	0.06	0.5	0.03–8	1	2	1–2
MRSP (*n* = 52)	ENR	32	64	0.06–64	64	≥512	0.125– ≥512	2	32	1–64
	MAR	32	64	0.25–64	64	64	0.5–64	1	2	1–2
	PRA	2	4	0.03–8	4	4	0.06–16	1	2	1–2
	CIP	32	32	0.125–128	256	512	0.125– ≥512	8	16	1–64
	MOX	4	8	0.06–8	0.06	8	0.125–32	2	2	1–8
*E. coli* (*n* = 65)	ENR	0.06	128	0.016–512	0.06	256	0.016–512	1	2	1–4
	MAR	0.06	64	0.016–256	0.125	64	0.016–512	1	2	1–8
	PRA	0.03	32	0.016–128	0.03	32	0.016–128	1	2	1–4
	CIP	0.03	256	0.008– ≥512	0.03	256	0.008– ≥512	1	2	1–8
	MOX	0.125	64	0.016–512	0.125	64	0.016–512	1	2	1–4

* Bactericidal effect if MBC/MIC ratio ≤ 4.

**Table 2 antibiotics-11-01204-t002:** Molecular analysis of 18 selected FQ-resistant isolates. Comparison between chromosomal mutations on DNA gyrase (*gyrA*) and topoisomerase IV (*grlA*/*parC*), plasmid mediated quinolone resistance (PMQR), ratio between MIC of pradofloxacin and MIC with pradofloxacin + efflux pump inhibitor (MIC_epi_) in selected resistant isolates. The isolates were chosen based on their pradofloxacin susceptibility.

ISOLATE ID	MIC Enrofloxacin (µg/mL)	MIC Marbofloxacin (µg/mL)	MIC Pradofloxacin (µg/mL)	MIC Ciprofloxacin (µg/mL)	MIC Moxifloxacin (µg/mL)	*gyrA*	*grlA (parC)*	PMQR	MIC/MIC_epi_
MSSA B053	16	32	1	16	4	Ser84Leu	Ser80Phe	NA	1
MSSA B071	8	16	1	16	4	**No mutation**	**No mutation**	NA	16 *
MSSA B074	8	16	2	32	4	Ser84Leu	Ser80Phe	NA	1
MRSA A019	16	32	1	128	4	Ser84Leu	Ser80Phe	NA	1
MRSA A069	64	128	4	256	1	Ser84Leu, Gly90Cys	Ser80Phe	NA	1
MRSA A132	16	64	32	256	16	Ser84Leu, Gly90Cys	Ser80Phe, Glu84Lys	NA	1
MSSP 098	4	4	0.5	4	0.5	Glu88Gly	Ser80Arg	NA	1
MSSP 115	16	16	2	16	2	Ser84Leu	Asp84Asn	NA	1
MSSP 099	32	32	4	16	4	Ser84Leu	Ser80Ile	NA	2
MRSP 045	32	32	1	32	4	Ser84Leu	Ser80Ile	NA	1
MRSP 038	64	32	2	32	2	Ser84Leu	Ser80Ile	NA	1
MRSP 067	64	64	4	32	4	Ser84Leu	Ser80Ile	NA	2
*E. coli* 10L-2253	8	32	4	32	16	Ser83Leu, Asp87Asn	Ser80Ile, Ala108Val	*-*	4 *
*E. coli* 16L-4063	64	32	4	128	16	Ser83Leu, Asp87Asn	Ser80Ile, Glu84Val	aac-(6’)-lb-cr	2
*E. coli* 15L-3275	128	64	32	64	64	Ser83Leu, Asp87Asn	Ser80Ile	QnrB	4 *
*E. coli* 282305	128	64	32	64	64	Ser83Leu, Asp87Asn	Ser80Ile	QnrB	8 *
*E. coli* 13L-4865	256	128	64	256	128	Ser83Leu, Asp87Asn	Ser80Ile, Ala108Val	QnrS	8 *
*E. coli* 13L-5283	128	256	64	128	256	Ser83Leu, Asp87Asn	Ser80Ile	QnrS, aac-(6’)-lb-cr	8 *
*E. coli* 13L-6009	512	256	128	256	512	Ser83Leu, Asp87Asn	Ser80Ile	QnrB, QnrS, aac-(6’)-lb-cr	8 *

(*): Efflux pump overexpression (MIC/MIC_epi_ ≥ 4). Yellow colour Resistant, Green colour Susceptible, increased exposure (according to EUCAST) NA= not assessed.

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
