# Peer review of "Investigation of In Vitro Susceptibility and Resistance Mechanisms in Skin Pathogens: Perspectives for Fluoroquinolone Therapy in Canine Pyoderma"

_antibiotics, 2022, doi:10.3390/antibiotics11091204_

Round 1

Reviewer 1 Report

This article entitled “Investigation of in vitro susceptibility and resistance mechanisms in skin pathogens: perspectives for fluoroquinolone therapy in canine pyoderma” have certain merits, due to the importance of the increased incidence of FQ-resistance, which represents a risk to animal and public health

 However, there are a number of minor changes, which should be incorporated.

Line 50.   FQs have a broad spectrum of activity and are listed as second choice antimicrobials [18]. Their use should be prudently limited to clinical cases where first-line antibiotics have been ineffective …

 It would be interesting include an advise about the Fluoroquinolones are classified for EMA (European medicines Agency) as an antimicrobial of second choice (following the recommendation of Critically Important Antimicrobials (CIA) in the World Health Organization’s CIA list)

 https://www.ema.europa.eu/en/documents/report/categorisation-antibiotics-european-union-answer-request-european-commission-updating-scientific_en.pdf

 Line 139 2.3. Resistance mechanism detection

S. aureus: in five selected resistant isolates (3 MSSA and 3 MRSA)

 Please check data

Line 251 Material and methods

… were collected from Germany between 2005-2006 and 2010-2011, respectively. E.  coli (n=65) were collected from the UK (Royal Veterinary College and University of Liverpool) from 2010 and 2017.

There is a lot of difference in years and that implies differences in antibiotic pressure from different FQ. They should explain something in the discussion

Author Response

Dear Reviewer,

The authors would like to thank you for your time and feedback on our manuscript and provide the following responses addressing your comments:

Reviewer 1

Line 50.   FQs have a broad spectrum of activity and are listed as second choice antimicrobials [18]. Their use should be prudently limited to clinical cases where first-line antibiotics have been ineffective …

It would be interesting include an advise about the Fluoroquinolones are classified for EMA (European medicines Agency) as an antimicrobial of second choice (following the recommendation of Critically Important Antimicrobials (CIA) in the World Health Organization’s CIA list)

 https://www.ema.europa.eu/en/documents/report/categorisation-antibiotics-european-union-answer-request-european-commission-updating-scientific_en.pdf

The information suggested have been included in the introduction.

 Line 139 2.3. Resistance mechanism detection

  1. aureus: in fiveselected resistant isolates (3 MSSA and 3 MRSA)

 Please check data

This has been amended.

Line 251 Material and methods

… were collected from Germany between 2005-2006 and 2010-2011, respectively. E.  coli (n=65) were collected from the UK (Royal Veterinary College and University of Liverpool) from 2010 and 2017.

Thank you. This information has been clarified for interesting future analysis in the discussion.

Kind regards

Stefano Azzariti

Reviewer 2 Report

This study described the in vitro susceptibility of several bacterial strains against fluroquionolones commonly used to treat skin infections in dogs. The study identified an alarming increase in ciprofloxacin resistance (512-fold increase in MBC90) toward MRSA and MRSP. Moreover, the authors studied resistance mechanisms and found mutations in bacterial topoisomerases along with efflux pump overexpression (in E. coli) and plasmid-mediated quinolone resistance.

The manuscript is well written, and the quality of presentation is good. In my opinion, the manuscript should be published in Antibiotics after addressing the following point.

1)      Since the authors detected several mutations in bacterial topoisomerases (gyrA, glrA, parC), the manuscript should discuss the corresponding changes in secondary structure of the protein and how it modifies FQ binding. Please explain the location of the mutated residues, if they are part of FQ binding sites etc. This can be done by analyzing the Protein Data Bank and some previous studies. Also, in silico tools such as molecular docking of FQs into the native and mutated protein could support the experimental data.  

Author Response

Dear Reviewer,

The authors would like to thank you for your time and feedback on our manuscript and provide the following responses addressing your comments:

This study described the in vitro susceptibility of several bacterial strains against fluroquinolones commonly used to treat skin infections in dogs. The study identified an alarming increase in ciprofloxacin resistance (512-fold increase in MBC90) toward MRSA and MRSP. Moreover, the authors studied resistance mechanisms and found mutations in bacterial topoisomerases along with efflux pump overexpression (in E. coli) and plasmid-mediated quinolone resistance.

The manuscript is well written, and the quality of presentation is good. In my opinion, the manuscript should be published in Antibiotics after addressing the following point.

1)    Since the authors detected several mutations in bacterial topoisomerases (gyrA, glrA, parC), the manuscript should discuss the corresponding changes in secondary structure of the protein and how it modifies FQ binding. Please explain the location of the mutated residues, if they are part of FQ binding sites etc. This can be done by analyzing the Protein Data Bank and some previous studies. Also, in silico tools such as molecular docking of FQs into the native and mutated protein could support the experimental data.

Thank you for the interesting comment. The information asked for have been revised with more details. The in silico analysis mentioned is intended to be done in future projects with an increased number of isolates, looking at the difference in FQ-binding activity between older and new generation FQs.

Kind regards 

Stefano Azzariti

Reviewer 3 Report

In this manuscript, the authors describe the comparison of MICs and MBCs distributions of specific veterinary and human antimicrobial fluoroquinolones in S. pseudointermedius, S. aureus and E. coli isolates exclusicely derived from canine pyoderma, and not just of canine origin. 

The manuscript is well-written, well-argued and has substantial clinical impact. The literature review is thorough.The methodology along with the results of the research are presented explicitly. The concept and the findings of this study should hold the attention of small animal practitioners as well veterinary microbiologists. In conclusion, this meticulous research work should be published.

Author Response

Dear Reviewer,

The authors would like to thank you for your time and feedback on our manuscript. Attached the revised draft. 

Kind regards

Stefano
